# Revolutionising the Quality of Life: The Role of Real-Time Sensing in Smart Cities

**Rui Miranda** [1] , **Carlos Alves** [1] , **Regina Sousa** [1] , **António Chaves** [1] , **Larissa Montenegro** [1] ,
**Hugo Peixoto** [1] , **Dalila Durães** [1,*] , **Ricardo Machado** [2] , **António Abelha** [1] , **Paulo Novais** [1]
and **José Machado** [1,*]

1 Algoritmi Research Centre/LASI, University of Minho, 4800-058 Guimarães, Portugal;
rui.miranda@algoritmi.uminho.pt (R.M.); carlos.alves@algoritmi.uminho.pt (C.A.);
regina.sousa@algoritmi.uminho.pt (R.S.); antonio.chaves@algoritmi.uminho.pt (A.C.);
larissa.montenegro@algoritmi.uminho.pt (L.M.); hpeixoto@di.uminho.pt (H.P.); abelha@di.uminho.pt (A.A.);
pjon@di.uminho.pt (P.N.)
2 Câmara Municipal de Guimarães, 4804-534 Guimarães, Portugal; ricardo.machado@cm-guimaraes.pt
* Correspondence: dad@di.uminho.pt (D.D.); jmac@di.uminho.pt (J.M.)

**Abstract:** To further evolve urban quality of life, this paper explores the potential of crowdsensing and crowdsourcing in the context of smart cities. To aid urban planners and residents in understanding the nuances of day-to-day urban dynamics, we actively pursue the improvement of data visualisation tools that can adapt to changing conditions. An architecture was created and implemented that ensures secure and easy connectivity between various sources, such as a network of Internet of Things (IoT) devices, to merge with crowdsensing data and use them efficiently. In addition, we expanded the scope of our study to include the development of mobile and online applications, emphasizing the integration of autonomous and geo-surveillance. The main findings highlight the importance of sensor data in urban knowledge. Their incorporation via Tepresentational State Transfer (REST) Application Programming Interface (APIs) improves data access and informed decision-making, and dynamic data visualisation provides better insights. The geofencing of the application encourages community participation in urban planning and resource allocation, supporting sustainable urban innovation.

**Keywords:** smart cities; crowdsensing; geofencing; data visualisation; mobile applications





## 1. Introduction

A smart city is an urban environment that integrates Information and Communication Technology (ICT) and the Internet of Things (IoT) to enhance the quality of life for its inhabitants, optimise resource efficiency, and improve overall sustainability through interconnected infrastructure, data-driven decision-making, and citizen engagement [1]. Urban centres increasingly depend on IoT and its expanding domains, as they attract the attention of academia, industry, and civil society [2]. Today, the main focus is on promoting citizens' quality of life, assessing the impact of intelligent technologies, and ensuring the social, economic, and environmental sustainability of cities [3]. At the heart of this urban transformation are two fundamental technological components: crowdsourcing and crowdsensing. Crowdsourcing allows citizens to participate actively in collecting data and evaluating services. At the same time, crowdsensing takes advantage of the ubiquity of sensors and mobile devices to facilitate the exchange of large amounts of valuable information in urban environments.

Active community participation is essential in developing plans for managing municipalities [4]. Despite the advances in sensorization and the use of mobile devices, some urban areas do not fully exploit these technologies. These devices could be interconnected by implementing a comprehensive platform that provides diverse solutions operating

at multiple levels and dimensions. The level of community involvement in the process is correlated with effectiveness in addressing relevant community concerns. Research results indicate that communities with abundant physical resources, highly skilled human capital, and strong social networks demonstrate greater competence when encouraged and included in the process [4].

This work followed the Design Science Research (DSR) methodology [5]. DSR is commonly used in information systems and related fields. It involves developing and evaluating novel artefacts to address identified issues or enhance existing procedures. This approach contrasts with other research paradigms, such as behavioral science, which seek to understand and explain events through observation and analysis [5]. The DSR process included the following steps: problem identification, aimed at providing a clear and concise definition of the problem; objectives definition, outlining the specific goals and outcomes sought in resolving the problem; design and development, involving the creation of a platform to solve the identified problem; demonstration, presenting the platform and its practical application in a tangible context; evaluation, conducting assessments to evaluate the platform's (artefact's) effectiveness in addressing the problem using empirical methodologies, simulations, or case studies; communication, presenting the findings, namely in this manuscript, among others.

Along with DSR, this article also implements a systematic literature review on the implementation of crowdsensing and geofencing technologies in smart cities [6]. This review aimed to identify the purpose, strategies, and tools using relevant articles from the Scopus database. Overall, these steps are intended to aid in the development of an innovative platform for the seamless integration of sensing technologies into the complex framework of smart cities, supported by solid knowledge and background. By merging crowdsourcing and crowdsensing, it could be possible to redefine the operational dynamics of urban centres, effectively meeting the needs of their residents with unparalleled efficiency [7].

This work uses crowdsensing and crowdsourcing, in which individuals actively contribute data via their networked devices, to provide a real-time view of the urban environment [8]. We propose and test an architecture to build a robust communication framework that efficiently collects data from various sources within the IoT ecosystem and across various sensing devices. This architecture ensures an efficient and secure data flow between these different sources.

The fundamental basis of our work lies in developing interactive dashboards, which allow urban planners and citizens to visually represent and understand the extensive data collection created by IoT [2]. This visualisation not only improves the effectiveness of decision-making processes but also facilitates the development of a deeper understanding of urban dynamics [9]. In addition, a smartphone application has been created that integrates autonomous geofencing functionality using data collected through crowd-sensing. Through crowd-sourcing, our app allows users to actively participate in establishing geographical boundaries and improving resource allocation. This research is a groundbreaking strategy for urban life in smart cities owing to its diverse influence on urban living, planning, and governance dynamics. The main characteristics this platform exhibits are:

- Integration of technologies: The platform establishes an extensive network of data points by integrating crowdsensing, crowdsourcing, and IoT technologies [10]. Crowd-sensing enables the acquisition of huge quantities of data from many individuals, effectively using the collective capabilities of citizen sensors. When combined with the accuracy of IoT devices, these data become further enhanced in terms of their robustness and reliability [11]. The use of an integrated approach facilitates comprehension of urban surroundings, hence potentially revolutionising the field of city planning and operations.
- Real-time responsiveness: One of the primary functionalities of the platform is its capacity to effectively analyse and present data in real time. The ability to respond promptly is of utmost importance in urban environments, where circumstances undergo rapid transformations, and the timely dissemination of information can sub-

stantially impact the overall well-being of individuals [3]. For example, real-time traffic and pollution data can enable individuals to make more informed choices regarding their travel routes and outdoor engagements, fostering a lifestyle promoting improved well-being.

- Enhanced citizen engagement: The prioritisation of mobile and online applications that incorporate geofencing and notification functionalities aims to enhance citizen participation. By engaging inhabitants in data gathering and dissemination, the platform improves the sense of communal belonging [12]. Citizens who possess knowledge and are actively involved are more willing to participate in municipal governance and projects, thus enhancing the democratic framework of the city.

- Data-driven decision-making: The usage and examination of data via this platform facilitate decision-making based on empirical evidence. City planners and politicians can discern recurring patterns, generate anticipatory projections, and optimise resource allocation more effectively [8]. For instance, identifying regions experiencing significant traffic congestion can facilitate the implementation of specific infrastructure modifications. In contrast, comprehension of utility consumption patterns can enhance the efficacy of energy management strategies.

- Sustainability and quality of life improvements: The knowledge obtained from the platform's data has the potential to drive sustainability endeavors. By monitoring environmental factors, urban areas can devise strategies to mitigate carbon footprints, effectively manage waste, and enhance the quality of green spaces. These actions directly impact the citizens' quality of life by promoting the development of a healthier and more sustainable urban environment [13].

The platform embodies a paradigm-shifting methodology as it engenders a change in urban existence, fostering enhanced connectivity, responsiveness, and sustainability. This framework offers a pragmatic approach to using technology to address the many issues associated with modern urban life. Its ultimate goal is to enhance smart cities' intelligence, sustainability, and livability. This study presents a vision of a future in which smart cities are interconnected and enhanced by their residents' collective intelligence. By integrating crowd detection, REST API architecture, data visualisation, and mobile application development, we aim to facilitate a paradigm shift in urban innovation and sustainability.

This manuscript is structured into six different sections. The article begins by introducing the need for innovative approaches in urban management and infrastructure development, emphasising the importance of the smart cities movement. It highlights the research focus on crowdsensing and crowdsourcing in the context of smart cities, underlining the potential to revolutionise the quality of urban life. Section 2 delves into core concepts, such as utilizing sensed data from a vast network of IoT devices and their integration with crowd-sourced data. This sets the stage for the study by discussing the architectural foundation for efficient data. Section 3 details the architecture created and implemented in the study to ensure safe and easy connectivity among diverse IoT data sources. It outlines the technical aspects of the architecture, highlighting its role in facilitating the integration of sensor data and crowd-sourced data. Section 4 presents the primary findings and showcases the critical role of sensor-collected data in enhancing urban knowledge and decision-making. It discusses the benefits of incorporating sensor data through REST API and the adaptability of cities through dynamic data visualisation. Section 5 offers a comprehensive analysis of the results, including the implications for city planning and the potential of dynamic data visualisation technology. It explores the role of the smartphone app's geofencing feature in promoting community involvement in urban planning and resource distribution. Finally, the manuscript summarises the key insights and the importance of sustainable urban innovation. It reiterates the goal of improving the standard of living and the flexibility of future cities through the collaborative use of expertise and cutting-edge technologies.

## 2. Background

In recent years, collective intelligence—characterised by rapid advances in digital technology—has emerged as a crucial factor in promoting creativity and solving complex problems. This section explores the fundamental principles and methodologies that form the basis of the dynamic domains of crowdsensing, crowdsourcing, API standards, data visualisation, and mobile applications. The interrelated subjects discussed have significantly influenced the methods by which data are collected, shared, and interpreted in our proposed architecture and platform, thus promoting the development of new ideas, cooperative efforts, and well-informed choices.

### 2.1. Crowdsensing and Crowdsourcing

Mobile crowdsensing has emerged as a highly favored approach to urban sensing, offering the ability to accumulate and disseminate significant amounts of data while monitoring and detecting the habits and movements of city dwellers in urban environments [14]. Crowdsensing is a method in which many individuals use mobile devices equipped with sensors to collaboratively share sensory data to quantify, examine, or deduce issues of common interest [7]. In the context of smart cities, crowdsourcing initiatives facilitate the active and passive participation of citizens in collecting data on various aspects of a city's operations, services, quality of life, and environment. Mobile crowdsourcing utilises the synergy between sensor technology and human input and analysis. This synergy proves especially valuable when assessing concerns related to the urban environment, such as the quality of municipal services and the ramifications of political decisions on residents' quality of life [15].

Shahrour et al. [15] state that crowdsourcing contributes to three critical success factors in smart city projects. The first factor concerns data acquisition, with mobile crowdsourcing enabling the development of cost-effective, high-caliber monitoring systems for urban infrastructures, services, and the environment. In addition, user feedback based on sensor-derived data is proving indispensable for capturing the needs and preferences of the population, as well as for understanding the genuine impact of smart city initiatives on resident's quality of life. The second factor revolves around involving citizens in local development and activities. Through mobile crowdsourcing, local authorities can access citizens' ideas and feelings about smart city initiatives and their tangible consequences. The third factor is the creation of smart applications supported by crowdsourcing, including, but not limited to, smart navigation, real-time public transportation services, ride-sharing arrangements, risk alerts, emergency responses, and disturbance notifications. By establishing cost-effective, crowdsourcing-driven monitoring systems as a viable alternative to traditional smart city monitoring devices, mobile crowdsourcing has the potential to accelerate the implementation of smart city projects. This research has identified a myriad of crowdsourcing methodologies applicable to smart cities. These cover the orchestration and management of crowdsensing campaigns and experiences, the collaborative selection of geographical boundary locations for bike sharing, emotional mapping to facilitate public discourse in shared spaces, the provision of information about the city and user satisfaction through mobile platforms, mobile crowdsensing with the ability to use extremities for potentially dangerous crowd situations, the use of social media data to monitor large-scale crowd events, the detection of on-street parking spaces through mobile crowdsourcing, and the evaluation of sound environments through crowdsourcing data. These methodologies use mobile devices, sensors, and community participation to collect and analyse data for smart city applications.

### 2.2. API Standards

There are several API standards used for various smart city applications, such as

1. Traffic data APIs: These APIs usually adhere to communication standards based on REST or the Simple Object Access Protocol (SOAP). They provide endpoints for accessing traffic data, such as information on congestion, accidents, and road

conditions, often in JavaScript Object Notation (JSON) or eXtended Markup Language (XML) format [16].

2.  Public transport APIs: Public transport APIs often follow open standards, such as the General Transit Feed Specification (GTFS), which defines a standard format for sharing information about public transport timetables, routes, and stops. These APIs often use REST to access real-time data, such as the location of vehicles [17].

3.  Parking APIs: Parking APIs may vary in their standards, but they often use REST to provide real-time information on the availability of parking spaces. They may adhere to secure communication protocols to guarantee data integrity [18].

4.  Air quality APIs: APIs that monitor air quality usually use REST communication protocols to provide up-to-date information on air quality at different locations in the city. Data can be transmitted in formats such as JSON or XML [13].

5.  Public lighting APIs: Public lighting or street lighting APIs can use communication standards such as Message Queuing Telemetry Transport (MQTT) to control lights based on light and motion sensors. This enables effective communication between sensors and control systems [19].

6.  Emergency service APIs: These APIs generally follow secure communication protocols like Hypertext Text Transfer Protocol Secure (HTTPS) to ensure the security of emergency-related information. They can provide real-time data about accidents, fires, and other critical situations [20].

In summary, the standards of APIs in smart city applications can vary. However, they often include REST, specific protocols for data types (e.g., GTFS for public transport), and security measures to protect sensitive information [9]. These standards ensure the interoperability and reliability of the APIs used to enhance efficiency and quality of life in smart cities. One significant area for improvement in this implementation is the absence of standardised practices in the sensitisation initiatives commonly used in smart cities, which is not uncommon, and to mitigate the absence of established criteria in the accessible data, the suggested framework incorporates a RESTful API to tackle this concern.

*2.3. Data Visualisation*

Data visualisation is paramount to comprehending, evaluating, and conveying the extensive and intricate datasets produced within smart cities; by data visualisation techniques, it is possible to process and present the data. Urban planning facilitates city development, land-use planning, and infrastructure optimisation. For public transportation, real-time transit data visualisation helps commuters with route planning, enhancing the overall efficiency of transit systems. Environmental monitoring leverages visualisations to track air quality, weather patterns, and energy consumption, bolstering sustainability initiatives [11,21,22]. In emergency response, geographic information system-based visualisations enable rapid, data-driven responses to disasters and emergencies. Moreover, data visualisation contributes to citizen engagement by empowering residents with insights and encouraging their participation in local governance. It further extends to gamification techniques to engage and inform the population effectively [12,23].

Significant progress has been made in data visualisation approaches to effectively handle the increasing complexity and volume of data in smart city contexts; such advances include the use of dashboards that offer real-time insights via interactive visual representations such as charts, maps, and interfaces [24]. By implementing such tools, policymakers and citizens can actively explore data while incorporating machine learning algorithms to assist in identifying patterns, anomalies, and correlations within extensive datasets. Consequently, it facilitates the decision-making process. Moreover, using data visualisation methods to depict information chronologically serves to augment comprehension of patterns, facilitate the ability to predict future events, and aid in improving municipal operations, among various other advantages [25].

A plethora of software tools and platforms have emerged to facilitate data visualisation in the context of smart cities. Tableau is widely used to construct dynamic dashboards

and reports, link disparate data sources, and provide real-time visualisation. Microsoft's Power BI software allows its customers to visualize data and access various business intelligence features. The software in question also facilitates effortless linkage to IoT data sources. QlikView and Qlik Sense are other applications with associative data modeling capabilities that let users explore data relationships interactively [26,27]. Furthermore, while pre-made solutions are prevalent, it is occasionally feasible to develop one's software by incorporating customisation and parameterization. Integrating a graphical library (ChartJS, LineJS, Google Charts, Leaflet) with a JavaScript framework (e.g., Angular, ReactJS, VueJS) is the prevailing method.

Despite the substantial advancements in data visualisation in smart city contexts, several persistent challenges continue to shape this field. These challenges include the delicate balance between data privacy and accessibility, the imperative for scalability to manage ever-expanding.

### 2.4. Mobile Apps

The development of crowdsensing involves integrating sensory data collection through users' mobile devices, allowing citizens to contribute to the collective through their devices. It provides personalised, intelligent information to each citizen, empowering them to make informed decisions and leverage community resources [28].

Previous literature on enhancing the quality of life within smart cities by incorporating crowdsensing, crowdsourcing, and geofences in mobile applications is evaluated. Devices and applications can use geofences to give helpful information to citizens when they are near an area of interest. The expansion of crowdsensing users and the development and implementation of smart cities share the objective of integrating citizens into intelligent environments via personalised information that is context-sensitive and transmitted via mobile applications.

According to Amaxilatis et al. [8], integrating IoT devices with smartphone applications is crucial for fostering and enhancing creative ecosystems in urban regions since this empowers users to verify and embrace new services. The authors concluded that the pervasive and routine smartphone usage among individuals provides a substantial opportunity for researchers to collect detailed and perceptive observations of urban surroundings.

Fernandes et al. [10], developed a mobile device platform that enables users to access comprehensive information about the current and future state of the city. The platform also provides context-aware smart notifications with relevant information. The platform incorporates a gamification mechanism incorporating user perception, evaluation, and satisfaction with their city to encourage continued usage. Meanwhile, Foschini et al. [14] presented ParticipAct, a mobile crowdsensing platform that uses edge nodes to identify potential hazardous crowd scenarios. This platform is particularly useful in emergencies, such as the ongoing COVID-19 pandemic, as it allows individuals to avoid dangerous crowd situations and receive suggestions for safer routes or places. ParticipAct has users install a sensing client application on their smartphones, which sends the collected information to a centralised cloud server. Researchers and platform administrators create these targeted sensing campaigns.

In summary, smart cities utilise information and communication technology to maximize infrastructure, facilitate cooperation, and promote innovation, all while ensuring efficient services. Active citizen participation in crowdsensing solutions can improve emergency management, environmental monitoring, healthcare and well-being, e-commerce, and other facets of smart cities. Mobile crowdsensing is an excellent method for gathering data in smart cities. Several methods can be implemented to promote public engagement, such as monetary incentives, entertainment, and service.

### 3. Architecture

The proposed methodology involves several technologies to create a crowdsensing platform for smart cities. Overall, the architecture presented in Figure 1 enables the

implementation of a framework aimed at improving the quality of life for citizens using IoT devices and crowdsensing data. First, data are collected from various sensors strategically placed in Guimarães. These data are then analysed and classified to extract valuable insights and identify patterns. The insights obtained are then used to automatically create geofences in specific locations relevant to citizens, such as areas with high traffic congestion or areas prone to natural disasters. Thus, the platform comprises three main components: web services, dashboards, and mobile apps. Web services integrate data from the IoT ecosystem to feed the data visualisation and Mobile app in real time. The mobile app allows citizens to interact with the platform by exploring the city map and receiving notifications when they enter a geofence. Finally, the back-office is a web application that allows platform administrators to manage the geofences created.

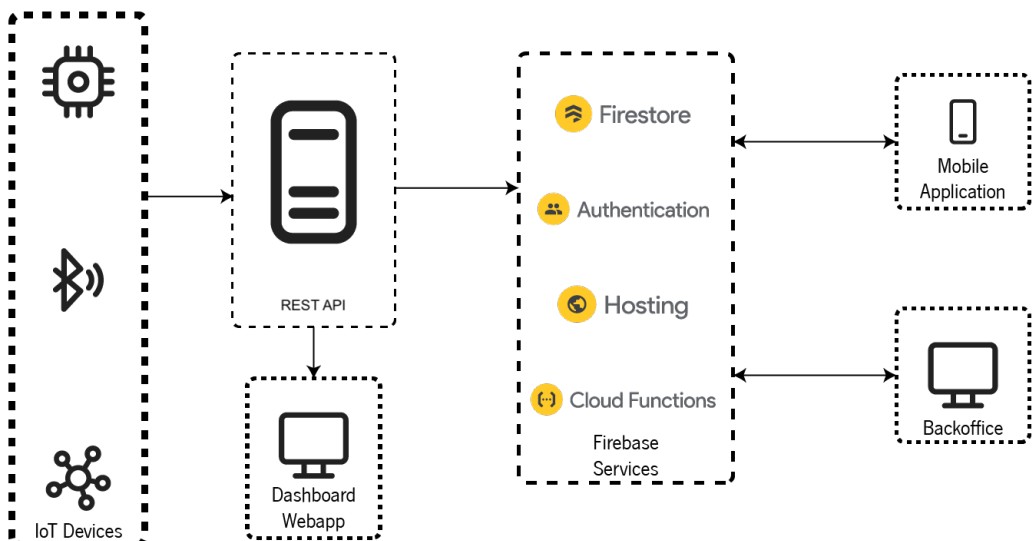

**Figure 1.** Overview of the platform's architecture.

Several tools and services from Firebase were used to develop these components. Firebase is a mobile and web development platform by Google that provides app developers with easy-to-use and scalable back-end tools and services. The tools and services used in this project include Cloud Functions, Cloud Firestore, Firebase Hosting, and Firebase Authentication. Cloud Firestore is a cloud-hosted NoSQL database that allows developers to store and sync data in real time across multiple clients and platforms. It was used to store the geofences created by administrators and external applications. Cloud Functions allow developers to run code responding to events triggered by Firebase features and HTTP requests. They were used to develop the RESTful web services that will enable external applications to interact with the platform. Firebase Hosting is a web hosting service that allows developers to deploy web apps and static content to a global content delivery network. It was used to host a back-office web application. Finally, Firebase Authentication was used in the back office, providing an easy-to-use and secure authentication system [29].

### 3.1. API

An API is a fundamental component in modern data collection services within urban environments. Specifically, within a city's data collection system, an API is the essential link that enables various software applications and systems to communicate and exchange data seamlessly [30].

The purpose of the designed API is to simplify the retrieval and sharing of data related to various aspects of urban life, including transportation, environmental quality, and municipal services. It empowers developers to programmatically access and manipulate the data, thereby enhancing the efficiency of data collection, analysis, and utilisation. APIs

developed for urban data collection provide a standardised set of functions and endpoints that grant authorised applications access to specific datasets. These APIs adhere to strict specifications and documentation to ensure consistency and reliability in data retrieval and transmission processes. Moreover, they are often compatible with multiple data formats, making them adaptable to various software systems. The API represents the intersection of technology and urban governance, optimising data-driven insights for the benefit of both city residents and administrators [31].

In our research, we developed the API in NodeJS, an open-source JavaScript framework. Our initial steps involved rigorous testing of external APIs for data collection to ensure the appropriate number of requests were made. Given the nature of the collected data, we determined that fetching data once daily would suffice for timely information. Equally important was the alignment of data types retrieved with each external API call. This information was subsequently organised into a proprietary ontology, enabling data storage in a MongoDB database, with separate documents for each type of reading and day. A similar process was applied to information available in text documents, which were parsed and stored according to the same rules.

How data were consumed by each service developed within the ecosystem influenced the definition of routing within the API. For instance, the web dashboard application benefited from mathematical operations applied to raw data, reducing the size of requests and visual information to enhance performance, necessitating the creation of specific routes for average or cumulative operations to ensure data remained relevant. Other services that did not require these routes could retrieve data in their raw states. We achieved deployment through Docker containers, which facilitated service management operations, including horizontal scaling, status management, and log persistence.

### 3.1.1. Dashboards Web App

The primary audience for the web application is urban corporations so that they can manage and monitor the data collected by the installed sensors in real time. The platform was built using ReactJS and NodeJS, with data from the API described in Section 3.1. The choice of ReactJS and NodeJS brings several benefits to the platform. Firstly, ReactJS ensures an efficient and responsive user interface, providing a smooth user experience, which is crucial for urban corporations that require real-time data monitoring. Additionally, NodeJS offers non-blocking I/O operations, which can significantly enhance the platform's overall performance, especially when dealing with real-time data. Furthermore, ReactJS and NodeJS are highly scalable technologies, capable of accommodating a growing number of sensors and data points over time. Their extensive developer communities provide access to a wealth of resources, libraries, and third-party modules, making it easier to address evolving needs and maintain the platform effectively. By accessing this platform, the offices responsible for administering and monitoring history and predictive analysis will discover three monitoring screens, each with a distinct function.

### 3.1.2. Mobile App

The designed mobile application enables users to interact with the created geofences. Users can explore the city map and receive notifications on their phones when they enter a geofence. This application, developed using React Native, supports the two most commonly used mobile operating systems—Android and iOS—using the same code, saving time and resources. Furthermore, it utilised the Expo platform, which offers a comprehensive suite of tools and services for React Native, to develop the application. We employed the InVision Studio screen design tool to create prototypes for the mobile application. InVision Studio allows teams to design, prototype, and animate within a unified application while utilising the InVision platform for seamless collaboration and teamwork.

## 4. Results

### 4.1. API Examples

The first example demonstrates a request to the external API for fetching sensor information. This request generates a list of all available sensors, providing relevant details about their type, GPS position, and the associated metrics.

```
GET /public/api/cways/latest/sensors HTTP/1.1
Host: https://api-sc.cm-guimaraes.pt/
Accept: */*
Content-Type: application/json
ApiKey: ---
```

Request line: The first line specifies the HTTP method, path and protocol version.
Host: Indicates the target server's hostname.
Accept: Lists the types of responses the client can handle.
Content type: Specifies the original media type of the resource.
API key: Secret authentication token.

The response, in JSON format, would include the following information, which was used to create various documents in the database. With recurring requests, values associated with each metric would be added to their respective documents, ensuring the retention of all pertinent information.

```
[
    {
        "name": "Fafe Norte",
        "id": "GUIM-FAFE-NORTE",
        "class": "mobility",
        "metrics": [
            "ligeirossaidacidade",
            "veiculossaidamais40km",
            "pesadossaidacidade",
            "peaonapassadeira"
        ],
        "latitude": 41.451333,
        "longitude": -8.280557
    },
    (...)
]
```

To obtain data from our API, the following request would provide a count of pedestrians crossing crosswalks in the city.

```
GET /reading/bydate?reading=peoes&start=1695312816954&end
    =1695313829193 HTTP/1.1
Accept: */*
Content-Type: application/json
ApiKey: ---
```

Request line: The first line specifies the HTTP method, path, query parameters, and protocol version.
Accept: Lists the types of responses the client can handle.
Content type: Specifies the original media type of the resource.
API key: Secret authentication token.

The following section illustrates an example response from the server. In this instance, the structured information is the raw data obtained from the database.

```
{
    "data": [
        {
            "_id": "63c55f78d49de618dc720cb5",
            "dataSourceUUID": "8326af69-dcd4-43b2-b732-8db73ba6b0c0",
            "readingType": "peoes",
            "category": "",
            "integrationType": "API",
            "values": [
                {
                    "date": "2023-09-21T17:32:03.060Z",
                    "value": 55
                },
                {
                    "date": "2023-09-21T17:32:54.171Z",
                    "value": 67
                },
                {
                    "date": "2023-09-21T17:33:25.634Z",
                    "value": 87
                }
                (...)
            }
        ]
    }
}
```

### 4.2. Dashboard Example

The web platform is divided into four significant dashboards.

Figure 2 illustrates the real-time data collected by sensors installed throughout the city. This display is divided into two main sections: weather and mobility data. To view current data, the user must select their desired location, and the data will be displayed in the various components corresponding to that location's coordinates.

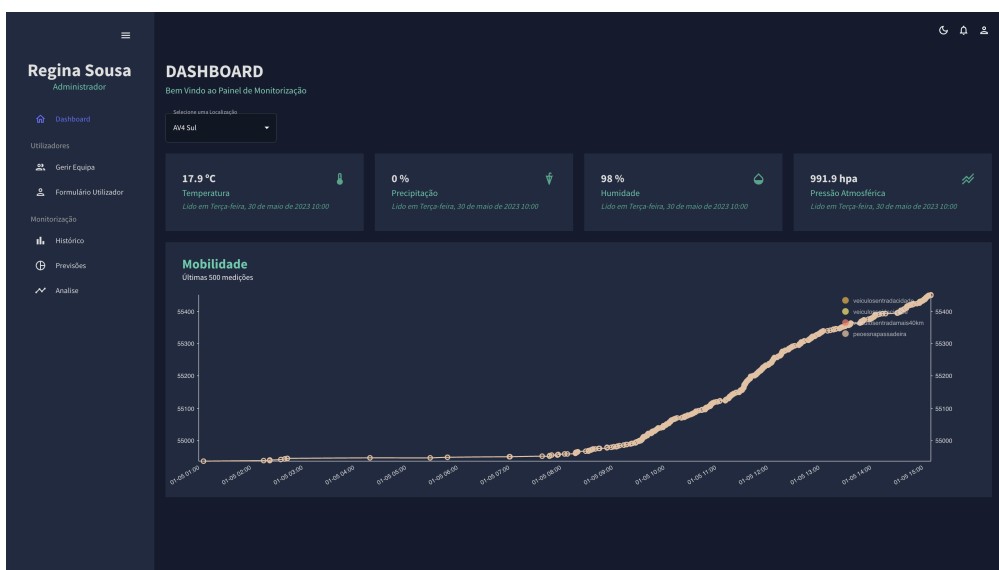

**Figure 2.** Dashboard tab with real-time data, collected by sensors for Mobility (Mobilidade).

In the second interface, as shown in Figure 3, historical data are accessible. It operates similarly to the previous interface, displaying data based on the location selected by the user.

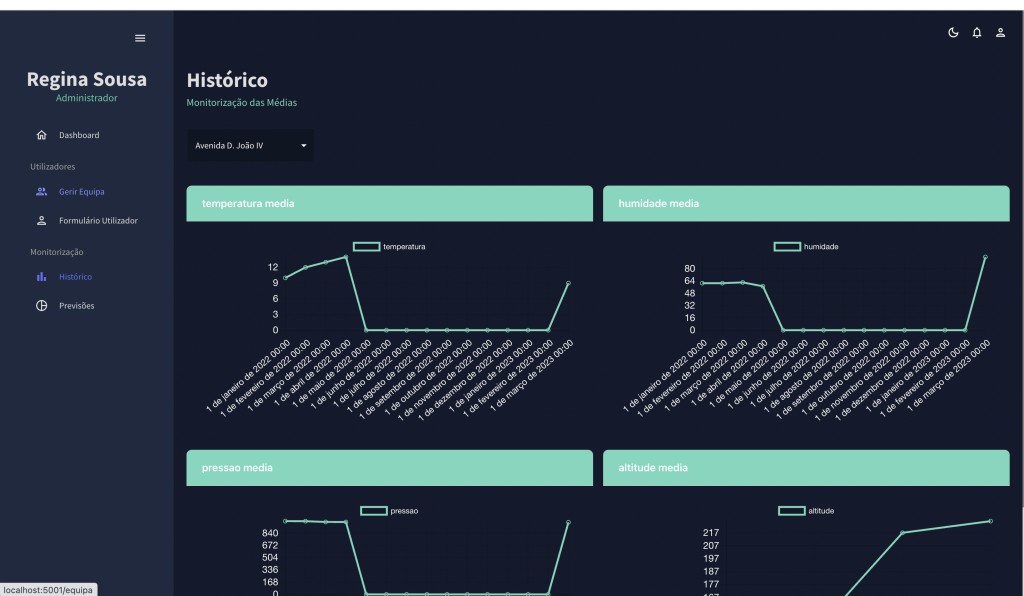

**Figure 3.** History data dashboard based on the selected street. The data are: Temperature, Humidity, Pressure, and Altitude.

Users are provided with two views: the isolated forecast data, as shown in Figure 4, and the comparative analysis, where historical data are presented alongside forecasts, as seen in Figure 5.

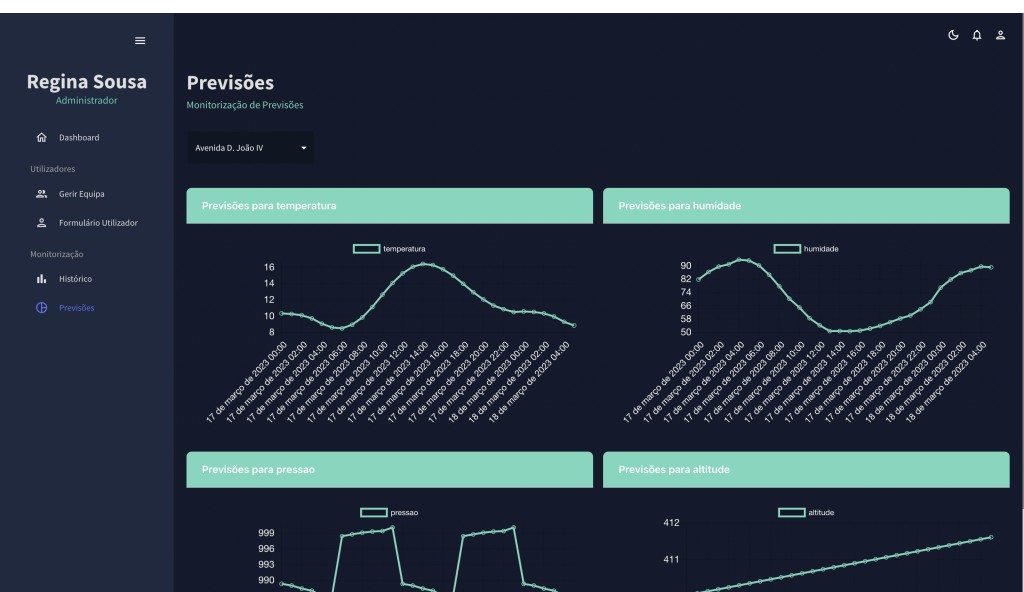

**Figure 4.** Result dashboard for forecasts with prediction for sensors with historical data. The data are: Temperature, Humidity, Pressure, and Altitude.

These dashboards are dynamic and scalable. In other words, only elements for which there are data are displayed. For instance, if only historical values for medical temperature exist for Street A, then only one graph will appear. If there are data from 10 sensors, 10 graphs will be displayed. If there are more than ten parameters, the graphs remain hidden until the user selects the graph title.

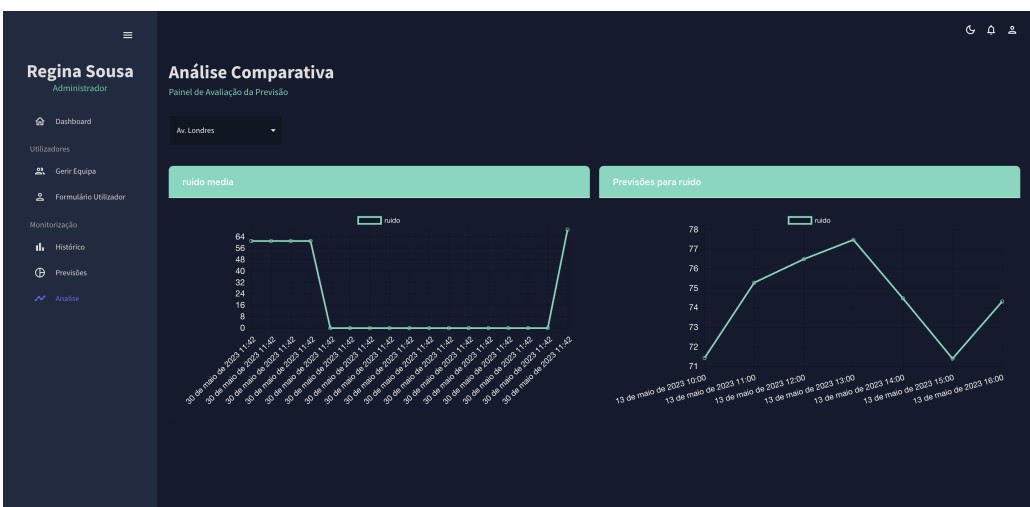

**Figure 5.** Comparative analysis with historical data and forecasts data for noise.

### 4.3. Mobile App

The mobile application is divided into components: onboarding, main screen, events, and settings. The onboarding component is the initial screen users encounter when they open the application, briefly introducing the platform and its key features. Figure 6 displays an example screen designed for Android users.

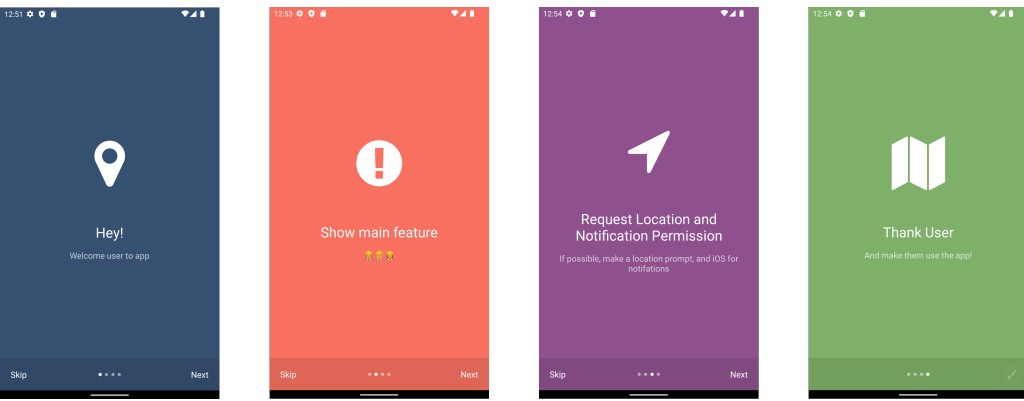

**Figure 6.** Onboarding interface running on an Android simulator.

The main screen is the central component of the application, displaying the city map and active geofences. This can be observed in Figure 7 for Android users and Figure 8 for iOS users.

Figure 7 shows Android users moving from left to right, and it shows the Android request to access the user's location, the main screen of the application with the city map and active geofences (with the map provided by Google Maps and the geofences retrieved from the Firestore database), and finally, the settings screen where the user can check if all location permissions are granted.

Figure 8 shows both the authorization screen and the general map. Additionally, the events component presents the user with information about events happening in the city, while the settings component allows users to customize the application's settings.

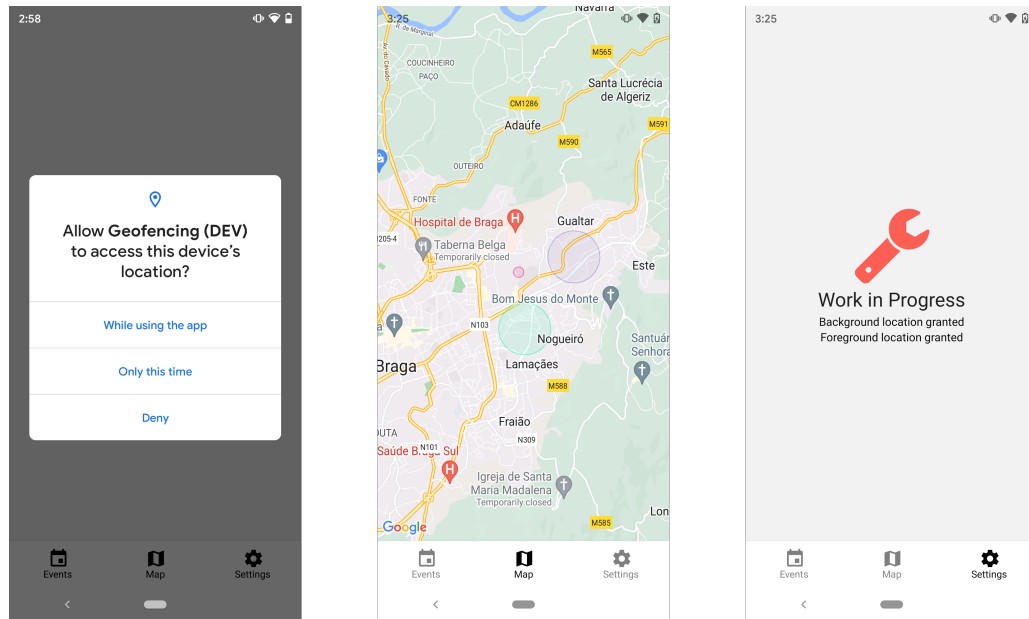

**Figure 7.** Application running on an Android device. This is the 3 step to view the geofences.

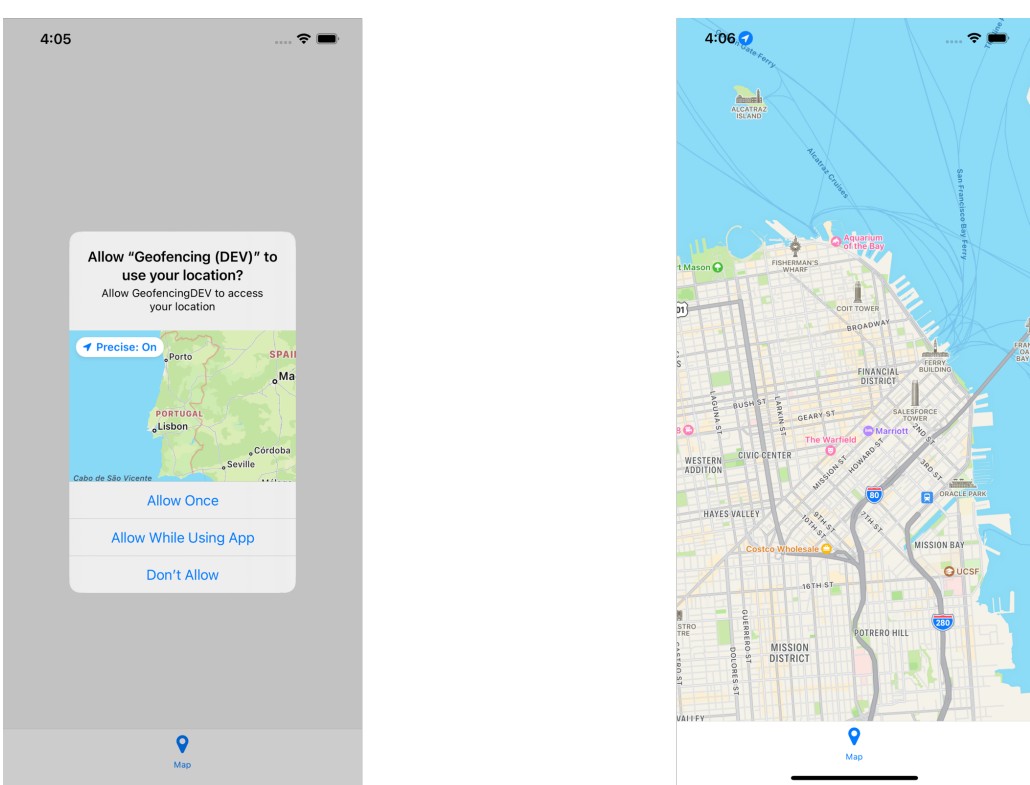

**Figure 8.** Application running on an iOS device. This is the 2 step to view the geofences.

Figure 9 displays the push notifications triggered when a user enters a geofence. Push notifications are activated upon entering and exiting the geofence location, ensuring that users are promptly informed about their surroundings.

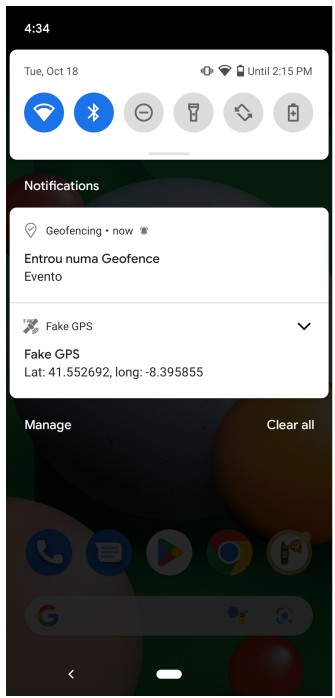
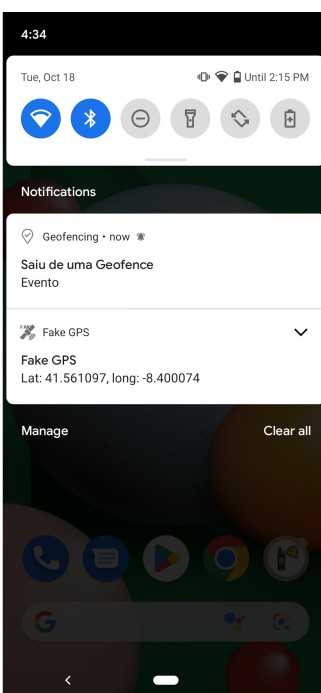

**Figure 9.** Notifications when an Android device enters or leaves a geofence.

## 5. Discussion

This project was a collaboration between the city of Guimarães and the University of Minho, with the goal of improving the quality of life for citizens by providing alerts and notifications relevant to their location. The first step in the project's development was identifying areas where sensors could be deployed to collect relevant data, followed by creating data collection and analysis processes to identify patterns and uncover insights.

Several contributions in the field of smart cities were achieved through this work. The study showcases a novel platform that exemplifies a promising path for advancing smart cities. It highlights the significant impact that can be achieved by integrating crowdsensing, crowdsourcing, and IoT technologies, emphasising their potential for transformation. The developed architecture establishes a foundation for a more networked and dynamic urban environment, wherein real-time data are central to urban planning and public involvement. The continual input of data from people and IoT devices forms a comprehensive and ever-changing representation of the city's activities.

An essential element of this system is its ability to promptly adapt to changing circumstances, although this capability is not without its inherent difficulties. The substantial quantity of data produced and the requirement for its immediate processing necessitate resilient computational resources and advanced algorithms. This requirement underscores the significance of ongoing investment in technology infrastructure and the advancement of sophisticated analytical tools. In addition, the effectiveness of real-time data systems is contingent upon the dependability and continuous operation of IoT devices and the network infrastructure. It is imperative to uphold these factors to avoid any interruptions in data transmission that may result in less-than-ideal decision-making.

While data were collected and analysed, work on a geofencing platform was already underway. The platform's goal was to automatically create geofences based on previous data and send relevant notifications to citizens. Notifications could include information about traffic congestion, available parking spaces, air pollution, and other city conditions.

Throughout the project's development, it was observed that real-time sensor data processing can provide new opportunities to smart cities and contribute to the development of a more sustainable and efficient urban environment in various ways, including monitoring and control, decision-making, improved public services, and citizen engagement. Cities can use real-time approaches to monitor and control multiple systems such as traffic,

energy, and water supply. Such information can be used to improve resource allocation, identify and resolve problems, and reduce waste and inefficiencies. A real-time analytic dashboard can also help optimise traffic management strategies by collecting data on traffic flow, congestion, and road conditions, identifying bottlenecks, recommending alternate routes, and providing drivers with real-time updates via intelligent notifications delivered to their smartphones, leading to less traffic congestion, shorter travel times, and increased road safety.

In the case of Guimarães, which aspires to be a smart city, a crowdsensing platform based on geofencing technology was proposed. The platform aimed to create geofences automatically based on previous insights and send relevant notifications to citizens. The platform can significantly improve citizens' quality of care and overall living experience by providing real-time information about traffic congestion, available parking spaces, and natural disasters.

## 6. Conclusions

Several contributions in the field of smart cities were made during this work. Thorough research into various topics, including crowdsensing, crowdsourcing, IoT, data visualisation, and mobile app development, yielded some intriguing results. Without a doubt, collaboration with a municipality and its real needs aided this work in evolving to solve and focus on real-world problems, transcending the boundaries of pure research. One of the first accomplishments of this work involved the collection of data from IoT devices provided by the city of Guimarães, as well as the implementation of data gathering and analysis procedures. These efforts resulted in discovering patterns and laid the groundwork for data-driven improvements in smart cities.

We consolidated several data sources and provided real-time information to several gateways, such as a mobile app, a web app with data visualisation features, and data storage for the application of future prediction models using a Restful API. The mobile app provides push notifications based on the user's geolocation. By using previous data analyses, geofences were generated automatically based on traffic events, cultural gatherings, and weather alerts. This platform has the potential to become an innovation hub, providing citizens with real-time information on critical events such as traffic congestion, parking availability, and air quality. This research demonstrates that real-time data analysis and the implementation of this crowdsensing technology can significantly improve the quality of life for Guimarães residents.

In retrospect, our findings confirm the importance of crowdsourced data and highlight the critical role of real-time sensor data processing in advancing more intelligent and environmentally friendly urban areas.

**Author Contributions:** Conceptualization, A.C., H.P., D.D., A.A. and P.N.; Methodology, R.S. and A.A.; Software, L.M. and R.M. (Ricardo Machado); Validation, P.N.; Formal analysis, R.M. (Rui Miranda); Investigation, R.M. (Rui Miranda), C.A., R.S. and L.M.; Resources, R.M. (Ricardo Machado); Data curation, P.N. and J.M.; Writing—original draft, R.M. (Rui Miranda), C.A., R.S., A.C. and L.M.; Writing—review & editing, H.P. and D.D.; Visualization, A.C. and H.P.; Supervision, D.D. and A.A.; Project administration, J.M.; Funding acquisition, J.M. All authors have read and agreed to the published version of the manuscript.

**Funding:** This work was supported by FCT-Fundação para a Ciência e Tecnologia within the R&D Units Project Scope: UIDB/00319/2020 and the project "Integrated and Innovative Solutions for the well-being of people in complex urban centers" within the Project Scope NORTE-01-0145-FEDER-000086. Rui Miranda was supported by grant no. UMINHO/BID/2021/137; Carlos Alves was supported by grant nos. 2022.12629.BD and UMINHO/BID/2021/134; Regina Sousa was supported by grant no. UMINHO/BID/2021/136; António Chaves was supported by grant no. UMINHO/BID/2021/135; Larissa Montenegro was supported by grant no. UMINHO/BID/2022/53.

**Data Availability Statement:** The data that support the findings of this study are available upon reasonable request from the corresponding author. The data are not publicly available due to restrictions imposed by the Municipality.

**Acknowledgments:** We would like to thank the Guimarães city hall for making the multiple datasets available.

**Conflicts of Interest:** The authors declare no conflict of interest.

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
