# Peer review of "Revolutionising the Quality of Life: The Role of Real-Time Sensing in Smart Cities"

_electronics, doi:10.3390/electronics13030550_

Round 1
Reviewer 1 Report
Comments and Suggestions for Authors
This study explores the transformative capacity of crowdsensing and crowdsourcing in the context of smart cities with the objective of revolutionizing the overall quality of urban life. The study defines and implements an overall architecture to effectively utilize sensed data, including crowd-sensed data, to advance dynamic data visualization tools, and to encompass the development of mobile and web applications. Ultimately, it provides a robust platform to promote urban innovation and sustainability
The topic of this study is interesting, as it explores approaches to improve the quality of life in smart cities by further discussing real-time sensing technologies based on the current status of cities. However, the contribution and the overall quality of the manuscript are not yet satisfactory in the current version. Several arguments and claims are expected to be elaborated more before the Results section. Please refer to the following points for revision.
1. Why and how does the "urban transformation lie two key technological components: crowdsourcing and crowdsensing"? There is a missing word in this sentence. Please verify.
2. The contributions of this study should be clearly stated in the Introduction. Currently, the Introduction seems duplicating the content of the Abstract without explicitly declaring the study's contributions. Except for the first paragraph, the following four paragraphs in the Introduction aim to assist readers in understanding how this manuscript is organized and the approaches adopted to "develop an innovative platform for the seamless integration of sensorization within the intricate framework of Smart Cities." The term "innovative" has not been well-defined, and it is often subjective. How and why can this innovative platform be considered an approach to revolutionize urban life in smart cities?
3. The last paragraph in section 2.2 provides some reflective thoughts, but other sections lack such insights for application in this study. There is also a typo in 'his project aim...'"
4. Section "Architecture" lacks citations entirely. Some content has also been mentioned in the previous section (Section "Background").
5. The section "Results" needs re-organization. There are no captions for Figures 2.1, 3.1, 4.1, and 5.1. It appears that Section 4.3 "Mobile App" contains more detailed demonstrations, potentially forming the main contributions of the study.
Overall, this manuscript is not yet ready for submission. The critical "Architecture" section lacks of clear indication of how it connects to the "Background" section. The "Results" section requires substantial revisions to clarify the study's contributions.
Comments on the Quality of English LanguagePlease review my comments to find where are those typos.
Author Response
Dear Reviewers,
The anwser is in attach file.

Reviewer 2 Report
Comments and Suggestions for Authors
The paper entitled "Revolutionising the Quality of Life: The Role of Real-Time Sensing in Smart Cities" proposes a novel framework designed to monitor life quality in the smart cities based on the IoT applications. The proposition is important, interesting and attractive. However, authors must do changes according to the following notes and remarks:
1- The Abstract is too long please rewrite it and just summarize important information (don't give detailed ones)
2- All abbreviations should only be used after their first definition (write each abbreviation in it full time when it is used for the first time; repeat that in the abstract and in the main text)
3- Add at the end of the introduction a chapter that summarizes how the rest of the paper is organized.
4- The text of some figures is not written in English (it is in Portuguese or Espana) (figures 2,3,4,5)
5- Add list of abbreviations in table in the introduction (not at the end of paper)
6- Why about 5 pages are reserved to the background section?
Please reorganize this section keep only important information
7- You should explain what the figure 1 illustrates
8- I can't check the obtained results illustrated in figures 2,3,4, and 5
9-Where is the related work section??
Author Response
Dear Reviewers,
The Answer is in attach file.

Reviewer 3 Report
Comments and Suggestions for Authors
1. The research purpose or hypothesis is unclear so it is difficult to judge whether your research framework and results are appropriate.
2. Your research method is unclear. Please clarify your research design, method according to your purpose.
3. It seems to be better to provide a more concrete background review about smart city as your introduction is too short. Further, for the application aspect, there are some literature you might be better to refer to:
Hsiao, H. (2021). ICT-mixed community participation model for development planning in a vulnerable sandbank community: Case study of the Eco Shezi Island Plan in Taipei City, Taiwan. International Journal of Disaster Risk Reduction, 58, Article 102218. https://doi.org/10.1016/j.ijdrr.2021.102218
T Kidokoro, R Fukuda, K Sho: GENTRIFICATION IN TOKYO: Formation of the Tokyo West Creative Industry Cluster, International Journal of Urban and Regional Research 46 (6), 1055-1077
Shigeru Takano, Maiya Hori, Yutaka Arakawa, Rin-ichiro Taniguchi, Towards ICT based mobility support system with in the COVID-19 era, In: The 18th ACM Conference on Embedded Networked Sensor Systems (SenSys 2020), 10.1145/3384419.3430609, 788-789, 2020.11.
4. Please revise the discussion section to clarify your main findings through comparing with findings from previous research based on the global context.
5. Totally, your literature review is insufficient. Please review some more paper including the introduction and discussion sections based on the international previous research.
Comments on the Quality of English LanguageModerate editing of English language required
Author Response

(The authors gave the same response as above.)

Round 2
Reviewer 3 Report
Comments and Suggestions for Authors
I appreciate the authors for their effort on revising the paper. Basically, all these revisions are considered satisfiable while there are some minor points might be improved before publication.
(1) You stated that “Smart cities represent a convergence of technology and infrastructures designed to sustainably and transparently improve lifestyles” in lines 19-20, while it will be better to briefly define “smart city” in the beginning considering both the academic and policy literature.
(2) Your literature review is still insufficient. For example, you mentioned the dynamics of urban centers in lines 58-59.
(3) Please fill in the right number of Figure ?? (maybe 6?) in line 480.
Comments on the Quality of English LanguageMinor editing of English language required before publication.
Author Response
All the answer are in attached file.
